# Cognitive Complaints in Patients with Suspected Obstructive Sleep Apnea Are Associated with Sleepiness, Fatigue, and Anxiety, Not with Final Diagnosis or Objective Cognitive Impairment

**DOI:** 10.3390/clockssleep7010012

**Published:** 2025-03-10

**Authors:** Tim J. A. Vaessen, Ruth E. Mark, Sebastiaan Overeem, Margriet M. Sitskoorn

**Affiliations:** 1Department of Psychiatry and Medical Psychology, Spaarne Gasthuis, 2000 AK Haarlem, The Netherlands; tvaessen@spaarnegasthuis.nl; 2Department of Cognitive Neuropsychology, Tilburg University, 5000 LE Tilburg, The Netherlands; r.e.mark@tilburguniversity.edu (R.E.M.); m.m.sitskoorn@tilburguniversity.edu (M.M.S.); 3Sleep Medicine Center “Kempenhaeghe”, 5590 AB Heeze, The Netherlands; 4Department of Electrical Engineering, Eindhoven University of Technology, 5600 MB Eindhoven, The Netherlands

**Keywords:** anxiety, cognition, cognitive complaints, obstructive sleep apnea, sleepiness

## Abstract

This study examined the nature, severity, and predictors of cognitive complaints in patients referred for suspected obstructive sleep apnea (OSA). The sample included 127 patients classified as no OSA (AHI, apnea/hypopnea index < 5, N = 32), mild OSA (AHI 5–15, N = 46), moderate OSA (AHI 15–30, N = 25), or severe OSA (AHI > 30, N = 24), and 53 healthy controls (HCs), matched for age, sex, education, and IQ. Cognitive complaints were assessed using the Cognitive Failure Questionnaire (CFQ) and the Behavioral Rating Inventory of Executive Functioning Adult Version (BRIEF-A). Regression analyses examined predictors of cognitive complaints including AHI, sleepiness, anxiety, depression, fatigue, and neuropsychological performance. Compared to HCs, those with mild OSA reported more forgetfulness, distractibility, and working memory issues, while those with severe OSA reported more difficulties with initiative, both with large effect sizes. Cognitive complaints were linked to sleepiness, anxiety, and fatigue (ß’s 0.29–0.37), but not AHI or cognitive performance. Cognitive complaints were not specific to subjects with OSA but were also common among individuals with sleep complaints suspected for OSA. In conclusion, cognitive complaints were associated with anxiety, fatigue, and sleepiness rather than objective cognitive performance or impairment.

## 1. Introduction

Obstructive sleep apnea (OSA) is a common sleep disorder affecting millions worldwide [1], and is linked to cardiovascular and metabolic complications [2]. OSA is also associated with cognitive deficits [3], particularly in attention, memory, and executive functioning. Patients with OSA commonly exhibit mild cognitive impairment [4] and perform worse on neuropsychological tests [5]. Interestingly, research has shown that these cognitive impairments do not always align with cognitive complaints—defined as individuals’ subjective concerns about their everyday cognitive abilities [6,7,8]. Cognitive complaints are self-reported problems or concerns with memory, attention, or executive functioning, assessed through open questions in interviews or via questionnaires. While some OSA patients under-report these complaints [9,10], they remain clinically significant due to their strong association with reduced quality of life [8,11] and increased healthcare utilization [12]. Studies suggest that cognitive complaints, rather than objective impairments, are more closely tied to quality of life in various medical conditions [11].

Research on cognitive complaints in OSA remains limited, however. Studies suggest that OSA patients report more cognitive complaints than healthy controls [13,14], particularly related to attention and memory [15]. However, findings on executive functioning complaints remain inconsistent [6]. Moreover, cognitive complaints do not appear to correlate with severity of OSA, as measured by the apnea–hypopnea index (AHI) [13], suggesting other contributing factors.

Potential influences on cognitive complaints in OSA remain unclear as well. Subjective sleepiness, measured by tools such as the Epworth Sleepiness Scale (ESS), has been proposed as a contributing factor [6], but other aspects such as fatigue [16], anxiety [17], and depression [18] have received less attention. They may contribute to actual cognitive difficulties or heighten patients’ awareness of cognitive lapses [19]. Given that fatigue, anxiety, and depression are prevalent in OSA [20], their role in cognitive complaints warrants further investigation. Understanding these relationships may help clinicians better address patient concerns and improve treatment strategies.

This study examined cognitive complaints in individuals referred to a sleep clinic for suspected OSA. We used validated questionnaires (Cognitive Failure Questionnaire, CFQ, and Behavior Rating Inventory for Executive Functioning Adult Version, BRIEF-A) to assess cognitive complaints and their potential correlates. Patients underwent overnight polysomnography (PSG) at a sleep laboratory to determine OSA diagnosis and severity (no OSA, mild, moderate, or severe), and their cognitive complaints were compared to a healthy control group recruited through hospital staff. Additional assessments included sleepiness (Epworth Sleepiness Scale, ESS), fatigue (Fatigue Assessment Scale, FAS), anxiety and depression (Hospital Anxiety and Depression Scale, HADS), and objective cognitive performance (paper-and-pencil and computerized tests from Central Nervous System Vital Sign, CNS-VS). We compared cognitive complaints between patients with and without an OSA diagnosis, as well as between cognitively impaired and unimpaired patients across OSA severity levels.

We hypothesized that patients referred for suspected OSA would report more cognitive complaints than healthy controls, regardless of OSA severity [15]. Based on previous research, we expected cognitive complaints to correlate with sleepiness [6], fatigue [16], anxiety [17], and depression [18] but not with objective cognitive performance [6,7,8]. Additionally, we predicted that cognitive complaints would not differ significantly between patients with and without a confirmed OSA diagnosis, as symptoms like sleepiness, fatigue, and anxiety can occur in those without OSA [21]. Lastly, we anticipated that cognitive complaints would not be linked to cognitive impairments across OSA severity levels [7].

## 2. Results

### 2.1. Participants

Figure 1 shows the flow chart for inclusion of patients suspected of OSA and healthy controls. We initially included 147 patients, but 20 had missing values on one or more measures. Those who dropped out were more likely to be female (*p* < 0.001); no differences were observed in AHI, age, education, or estimated IQ (all *p*’s > 0.29). The remaining 127 were categorized as having no OSA (AHI < 5, N = 32), mild OSA (AHI 5–15, N = 46), moderate OSA (AHI 15–30, N = 25) and severe OSA (AHI > 30, N = 24). Among those without OSA, two were diagnosed with restless legs syndrome, one with periodic limb movement disorder, and one with both snoring and insomnia. Four had a history of a burn-out, one had panic attacks, and one had a leaky heart valve. The remaining 24 had no significant medical or psychiatric history. We recruited 53 healthy controls, excluding 5 to match the OSA groups on age, sex, education, and estimated IQ at group level.

### 2.2. Demographic Characteristics and Sleep Measures

Table 1 summarizes the demographic and sleep measures of the included groups. Patients and controls did not differ with respect to age, sex, education, and estimated IQ. (all *p*’s > 0.10). All patient groups displayed higher Pittsburg Sleep Quality Index (PSQI) scores than controls (all *p*’s < 0.001). Patients without OSA (AHI < 5) more often exceeded the PSQI clinical cut-off than other OSA groups and controls (all *p*’s < 0.01). Mild (AHI 5–15) and severe OSA (AHI > 30) patients scored higher on the ESS than controls (all *p*’s < 0.05). The prevalence of excessive sleepiness did not differ across OSA severity and from healthy controls, however (all *p*’s > 0.51).

### 2.3. Cognitive Complaints

Cognitive complaints were assessed using the CFQ and the BRIEF-A. The subscales of both questionnaires cover various complaints regarding attention, memory, and executive functioning. Table 2 presents CFQ and BRIEF-A subscale scores across OSA groups and controls. Patients with mild OSA (AHI 5–15) reported more complaints about forgetfulness, distractibility, and working memory than controls (all *p*’s < 0.03) with large effect sizes (all Cohen’s ds > 0.8). Patients with severe OSA reported more initiative problems (*p* < 0.001) with a large effect size (Cohen’s d = 1.1). Patients with no OSA (AHI < 5) or moderate OSA (AHI 15–30) did not differ from controls on any of the CFQ and BRIEF-A subscales (all *p*’s > 0.06).

### 2.4. Anxiety, Depression and Fatigue

Anxiety and depression were measured using the HADS anxiety and depression scales and fatigue was measured using the FAS scale, with results shown in Table 3. Patients with no or mild OSA reported more severe anxiety and depression symptoms than controls (all *p*’s < 0.04). All OSA groups reported more fatigue than controls (all *p*’s < 0.001). Anxiety, depression, and fatigue levels did not differ among OSA groups (all *p*’s > 0.11).

### 2.5. Neuropsychological Tests

Patients and controls underwent a neuropsychological assessment that covered the domains of attention, memory, and executive functioning. Table 3 presents norm-corrected Z-scores for neuropsychological tests for all groups. Patients without OSA performed worse on episodic memory (RAVLT immediate recall) compared to controls (*p* < 0.01). No other significant differences were found across groups (all *p*’s > 0.09).

### 2.6. Predictors of Cognitive Complaints in Suspected OSA

We performed multivariate regression analyses for all CFQ and BRIEF-A subscales (see Table 4 and Table 5). Models were significant (*p* < 0.04), except for BRIEF-A inhibit (*p* = 0.19) and BRIEF-A self-monitor (*p* = 0.67), with R^2^ values ranging from 0.23 to 0.42. Higher ESS scores predicted forgetfulness (ß = 0.32, *p* < 0.001), distractibility (ß = 0.31, *p* = 0.01), and false triggering (ß = 0.34, *p* < 0.001). Higher HADS anxiety scores predicted more issues with shifting (ß = 0.40, *p* < 0.001), emotional control (ß = 0.35, *p* = 0.01), and planning/organizing (ß = 0.29, *p* = 0.03). Higher FAS scores were linked to more forgetfulness (ß = 0.32, *p* = 0.01), problems with initiation (ß = 0.35, *p* < 0.001), working memory issues, (ß = 0.37, *p* < 0.001), and planning/organizing difficulties (ß = 0.29, *p* = 0.03).

### 2.7. Cognitive Complaints and OSA Diagnosis

Not only did we categorize patients in different groups based on the AHI, we also split patients into those who received an OSA diagnosis based on clinical criteria (see Section 4.1.1.) and those who did not meet clinical criteria for OSA. Of the 127 patients suspected of OSA, 89 (70%) received an OSA diagnosis based on clinical criteria. Table 6 presents total CFQ and BRIEF-A scores for patients with and without a clinical OSA diagnosis, and healthy controls. Patients without OSA reported more cognitive complaints on the CFQ than controls (*p* = 0.04). Those with an OSA diagnosis reported more complaints on the BRIEF-A than controls (*p* = 0.02). However, cognitive complaints did not significantly differ between patients with and without an OSA diagnosis (all *p*’s > 0.06).

### 2.8. Cognitive Complaints and Cognitive Impairments

Patients were categorized as cognitively impaired if the score of two or more neuropsychological tests fell below international criteria for cognitive impairments [22]. Cognitive impairment was identified in 34% of the subjects with no OSA, 53% of those with mild OSA, 56% of those with moderate OSA, and 25% of those with severe OSA. Table 7 shows the CFQ and BRIEF-A scores for cognitively impaired and unimpaired patients across severity levels. Within each OSA severity group, cognitive complaints did not differ between impaired and unimpaired patients (all *p*’s > 0.16).

## 3. Discussion

This study examined cognitive complaints in patients referred to a sleep clinic with suspected OSA. Compared to a healthy control group, patients with mild OSA (AHI 5–15) reported more frequent complaints related to forgetfulness, distractibility, and working memory, with large effect sizes. Patients with severe OSA (AHI > 30) reported greater difficulties with taking initiative, also with a large effect size. These findings expand the existing literature by highlighting that OSA patients not only report memory and attention-related issues but also complaints associated with executive functioning [6]. Furthermore, our results suggest that the nature of cognitive complaints may vary with OSA severity. Interestingly, cognitive complaints were also prevalent in individuals referred to the sleep clinic under suspicion of OSA, regardless of whether they received an OSA diagnosis. This suggests that cognitive complaints are not specific to OSA, but common among individuals with sleep complaints suspected of OSA.

Our findings indicate that subjective sleepiness, anxiety, and fatigue were the strongest predictors of cognitive complaints, independent of OSA severity. This aligns with previous research demonstrating a link between subjective sleepiness and cognitive complaints [6], as well as studies in other medical conditions that have highlighted the role of fatigue and anxiety in self-reported cognitive difficulties [16,17]. The cross-sectional design of our study limits causal conclusions, but bidirectional relationships may exist. Anxiety, fatigue, and sleep problems may contribute to cognitive complaints by impairing focus and increasing distractibility [23]. Anxiety also increases vigilance, leading to catastrophizing and over-reporting minor cognitive lapses [19,24]. Similarly, fatigue and sleep disturbances can exacerbate cognitive complaints by impairing working memory and attention [25,26]. Conversely, cognitive complaints may themselves contribute to increased stress, worry, and fatigue [27], perpetuating poor sleep and daytime dysfunction. Future longitudinal studies are needed to clarify these relationships.

Our results also showed that cognitive complaints in OSA are not directly related to cognitive performance or underlying cognitive impairments. Despite 25–53% of our patients being classified as cognitively impaired, these individuals were not more likely to report cognitive complaints. This finding is consistent with prior studies in OSA [6] and underscores the complexity of cognitive complaints. Clinicians should be aware that cognitively impaired patients may not report difficulties due to lack of insight, denial, or reduced awareness of deficits. Moreover, standard neuropsychological tests may not adequately capture daytime vigilance impairments, which are commonly observed in sleep disorders like OSA [28]. Traditional tests assess cognitive abilities under optimal conditions, potentially overlooking fluctuations in alertness and attention deficits caused by sleep disruption [28,29].

A key strength of this study is the use of validated questionnaires to assess cognitive complaints across multiple domains, along with the inclusion of patients across the full spectrum of OSA severity. By comparing OSA patients with both healthy controls and sleep center patients without OSA, we gained two important insights. First, cognitive complaints can be present even in mild OSA and in individuals without a formal OSA diagnosis. Second, non-OSA-specific factors, such as fatigue, anxiety, and subjective sleepiness, play a significant role in cognitive complaints, emphasizing the need to assess these symptoms alongside OSA severity.

This study has limitations as well. First, we only assessed AHI as a measure of OSA severity. Emerging research suggests that OSA is a heterogeneous condition with phenotypic subtypes, characterized by varying degrees of symptoms such as sleep fragmentation, hypoxemia, and daytime sleepiness [29]. These phenotypes may be linked to different patterns of cognitive dysfunction [30]. While prior studies have explored phenotypic differences in objective cognitive impairment [31], no research has yet investigated whether subjective cognitive complaints differ between OSA phenotypes. Understanding these differences could provide insight into why some patients experience pronounced cognitive complaints despite minimal objective deficits, whereas others with cognitive impairments report few complaints. Large cohort studies are needed to explore these associations, as our sample size was too small for such an analysis. Future research should examine whether distinct OSA phenotypes are associated with specific patterns of cognitive complaints and whether this could inform personalized treatment approaches.

Second, dropout analyses revealed that women were more likely to discontinue participation, potentially limiting the generalizability of our findings to female patients. There is evidence suggesting that women with medical diseases may be more prone to cognitive complaints than men [16], making gender-specific analyses an important area for future research. Additionally, our sample was predominantly white European, limiting the applicability of findings to more diverse populations. Prior research has indicated that predictors of cognitive complaints may vary across racial and ethnic groups, highlighting the need for studies with more diverse participant samples and limiting the generalizability of findings. Prior research has indicated that predictors of cognitive complaints may vary across racial and ethnic groups [32]

This study has important clinical implications. At sleep clinics, clinicians should be cautious when attributing cognitive complaints solely to OSA or to cognitive impairments, especially in patients without neurological or psychiatric comorbidities. Our findings suggest that cognitive complaints in this population are more strongly linked to symptoms such as fatigue, anxiety, or other sleep problems rather than OSA itself. When patients report cognitive complaints, additional evaluations for coexisting sleep problems (e.g., insomnia), anxiety, or fatigue-related conditions should be considered. Treatment strategies targeting these symptoms, such as cognitive behavioral therapy for insomnia or psychological interventions for anxiety and fatigue, may be beneficial.

Future studies should focus on the generalizability of the findings by including more women and participants from diverse racial and ethnic backgrounds. Moreover, studies should focus on OSA patients outside sleep centers, such as those identified through general community screening, or those with neurological or psychiatric comorbidities, to test the generalizability of our findings. Another crucial area of future research is to investigate the effect of OSA treatment on cognitive complaints. While OSA treatments, such as CPAP or mandibular advancement devices, are known to improve sleepiness [33] and mood symptoms [34], their impact on cognitive complaints remains poorly understood. Given the links between cognitive complaints, sleepiness, and mood disturbances, it is plausible that OSA treatment could alleviate cognitive complaints as well. Randomized controlled trials (RCTs) assessing the effects of OSA treatment on cognitive complaints, particularly in relation to subjective sleepiness and mood symptoms, are needed to clarify these potential benefits.

## 4. Materials and Methods

### 4.1. Participants

#### 4.1.1. Patients with Suspected OSA

Participants were referred by their general practitioner to sleep units at VieCuri Medical Center (Venlo) and Reinier de Graaf Hospital (Delft) for suspected OSA. Referral was based on self-reported symptoms such as snoring and observed apneas. Patients were recruited before undergoing overnight PSG. Based on the AHI, they were categorized as follows: no OSA (AHI < 5), mild OSA (AHI 5–15), moderate OSA (15–30), and severe OSA (AHI > 30). Final OSA diagnosis followed the American Academy of Sleep Medicine guidelines [35]: AHI of  ≥15 or AHI of  ≥5 with significant daytime symptoms, such as an elevated Epworth Sleepiness Scale (ESS) score, requiring treatment. Inclusion criteria were as follows: 1. native Dutch speaker; 2. age 18–70 years old; 3. no substance or alcohol abuse (max. three units/day for males, two for females); 4. no psychiatric diagnoses; 5. no cognition-affecting medication; 6. no neurological diseases, diabetes mellitus, thyroid disease, or pulmonary disease. Screening was based on verbal report and medical records.

#### 4.1.2. Healthy Controls

Healthy controls were recruited from hospital personnel and their relatives. Inclusion criteria ensured a low risk of sleep disorders: 1. no snoring or apneas (confirmed by a partner); 2. PSQI score ≤ 7 [36]; and 3. ≥80% sleep efficiency based on a two-week sleep diary. The same six inclusion criteria as for suspected OSA patients were applied [37,38,39]. Controls did not undergo PSG.

The study followed the Declaration of Helsinki and was approved in March 2012 by a national ethics committee (NL37795.068.11, data collected from June 2012 until April 2018). Written informed consent was obtained.

### 4.2. Measures

#### 4.2.1. Demographic Characteristics

Data on age, sex, and education were collected. Educational level was rated on the Dutch Verhage scale [40] (comparable to the International Standard Classification of Education) [41]. Premorbid verbal intelligence quotient (IQ) was estimated using a Dutch reading test [42].

#### 4.2.2. Sleep Measures

Sleep quality was assessed with the Pittsburgh Sleep Quality Index (PSQI) [36], a validated and reliable self-report questionnaire consisting of 19 items and resulting in a total score ranging from 0 to 21. Higher scores indicate worse sleep quality [43,44]. Subjective sleepiness was measured with the Epworth Sleepiness Scale (ESS), where scores above >10 indicate excessive daytime sleepiness [45]. OSA severity was determined by the AHI and lowest nighttime oxygen saturation (SaO_2_lowest) from PSG.

#### 4.2.3. Cognitive Complaints

Cognitive complaints were assessed with the Cognitive Failure Questionnaire (CFQ) and the Behavior Rating Inventory of Executive Function-Adult (BRIEF-A). The CFQ measures cognitive lapses over three months (25 items, score range 25–125). Subscales assess forgetfulness, distractibility, and false triggering. BRIEF-A evaluates executive function complaints over the past month (75 items, score range 75–225) across nine subscales. Higher scores indicate greater impairment. For an explanation of the content of the subscales, see Table 8. As for missing data, if one item was missing on a subscale, it was replaced with the mean subscale score. If multiple items were missing, the subscale was excluded.

#### 4.2.4. Depression, Anxiety, and Fatigue

Mood symptoms were assessed with the Hospital Anxiety and Depression Scale (HADS), including anxiety (0–21) and depression (0–21) subscales. Fatigue was measured with the Fatigue Assessment Scale (FAS, 10 items, range 10–50). Missing data were handled as described for cognitive complaints.

#### 4.2.5. Neuropsychological Tests

A combination of paper-based and computerized tests assessed cognitive function.

Paper-and-Pencil Tests:Working memory: WAIS-III Digit Span total score [43].Episodic verbal memory: Rey Auditory Verbal Learning Test (immediate recall) [44].Divided attention: Trail Making Test AB index [46].Verbal fluency: Total correct responses in Animal Fluency [47].

Except for Digit Span, scores were standardized (Z-scores) for age, sex, and education.

Computerized Tests:CNS Vital Signs battery [48]: Symbol Digit Coding (processing speed), Stroop Test (reaction time), Shifting Attention Test (cognitive flexibility), Continuous Performance Test (complex attention).CNS-VS domain scores were calculated and standardized for age, sex, and education [49].

Cognitive impairment was classified using the Cognitive Impairment Non-Dementia (CIND) criteria [22]. A test score below −1.65 SD was considered impaired. Patients were classified as cognitively impaired if they had ≥2 impaired scores.

### 4.3. Statistics

A power analysis for multilevel analysis was performed following the method described by Twisk [50]. Assuming a high reliability of 0.70 [51,52], a significance level (alpha) of 0.05, and a power of 0.80, it was determined that 64 participants were needed in each group (OSA and HC) to detect a medium effect size (at least 0.5 standard deviation) on the total scores of the CFQ and BRIEF-A.

All OSA groups and healthy controls were matched at group level on age, sex, educational level, and estimated IQ using one-way analysis of variance (ANOVA) for continuous variables and Chi-square tests for categorical variables. Healthy controls that differed significantly from the OSA groups were excluded one-by-one until subjects matched across all groups. Demographic characteristics, sleep measures, cognitive complaints, anxiety, depression, fatigue, and performance on neuropsychological tests were compared between all groups using one-way analysis of variance (ANOVA) for continuous variables. If significant, post hoc independent t-tests were conducted. Chi-square tests were used for categorical variables. *p*-values were adjusted using the Benjamini–Hochberg method [53].

For all patients suspected of OSA, regression analyses were conducted to identify predictors of cognitive complaints (CFQ and BRIEF-A subscales). Predictors included OSA severity (AHI), sleepiness (ESS), anxiety (HADS), depression (HADS), fatigue (FAS), and the performance on each neuropsychological test.

We also compared healthy controls, patients with an OSA diagnosis, and patients without an OSA diagnosis using ANOVAs on the total scores of the CFQ and the BRIEF-A. If significant, post hoc *t*-tests were performed.

Within each OSA severity category, cognitively impaired and unimpaired patients were compared using independent t-tests on total CFQ and BRIEF-A scores.

Two sided *p*-values are reported and a *p*-value < 0.05 was considered statistically significant. *p*-values for the regression analysis were adjusted with the Benjamini–Hochberg method to correct for multiple testing [53]. Only corrected *p*-values are reported. All statistical analyses were performed using SPSS 24.0.0.0 for Windows.

## 5. Conclusions

Cognitive complaints are common among patients seeking evaluation for suspected OSA, regardless of the final diagnosis. In OSA patients, cognitive complaints were linked to subjective sleepiness, anxiety, and fatigue. However, the complaints did not reflect objective cognitive impairments. Patients can be reassured that their cognitive concerns are unlikely to indicate true cognitive deficits. When complaints persist, further evaluation and treatment of sleep disturbances, anxiety, and fatigue should be considered.

## Figures and Tables

**Figure 1 clockssleep-07-00012-f001:**
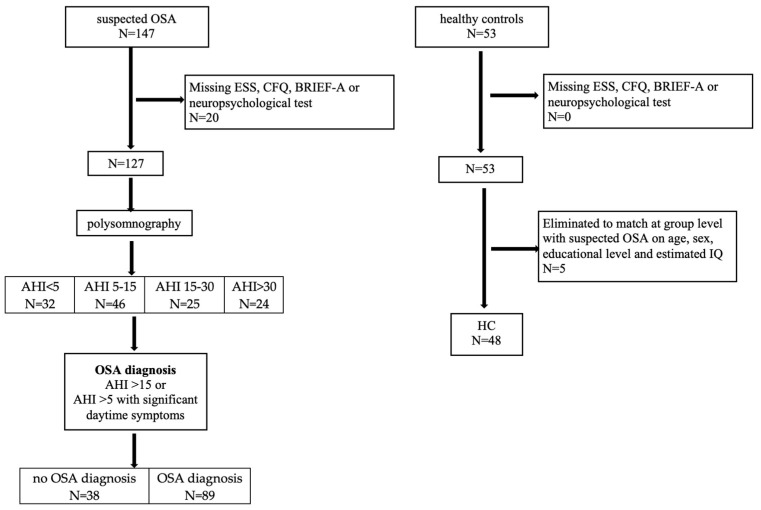
Flow chart suspected OSA and healthy controls.

**Table 1 clockssleep-07-00012-t001:** Demographic and sleep variables in patients with suspected OSA and healthy controls (HC).

	Range	HC N = 48	AHI < 5 N = 32	AHI 5–15 N = 46	AHI 15–30 N = 25	AHI > 30 N = 24	ANOVA/χ^2^			
							df	F	χ^2^	*p*
Demographics										
Sex (% male)	0–100	69%	72%	78%	80%	83%	4	-	2.66	0.62
Age	>18	52.9 (8.8)	46.8 (11.8)	49.4 (10.7)	53.3 (10.1)	49.8 (11.4)	4;170	2.31	-	0.10
Educational level	1–7	5.3 (0.7)	5.1 (0.8)	5.5 (1.0)	5.2 (1.2)	5.2 (1.2)	20	-	26.60	0.21
Estimated IQ	M = 100/SD = 15	97.6 (12.1)	91.9 (14.0)	94.9 (16.6)	98.0 (17.1)	94.5 (15.0)	4;170	0.94	-	0.55
Sleep variables										
PSQI	0–21	3.0 (1.6)	7.4 (3.4) ***	6.3 (2.8) ***	6.7 (3.6) ***	7.1 (3.7) ***	4;170	14.79	-	<0.001
PSQI > 8	0–100	0%	42% ***	30%	28%	36%	4	-	23.44	<0.001
ESS total score	0–24	4.5 (3.4)	7.1 (4.0)	6.9 (3.9) *	6.4 (4.0)	7.8 (3.9) **	4;170	4.11	-	<0.01
ESS > 10	0–100	8%	16%	12%	12%	21%	4	-	0.51	0.51
AHI	>0	na	2.5 (1.3) ***	8.8 (2.7) ***	21.7 (5.3) ***	50.4 (17.8) ***	3;123	182.68	-	<0.001
SaO_2_lowest	<100%	na	89.8 (2.0) ***	85.7 (4.2) ***	80.8 (6.6) ***	76.8 (7.1) ***	3;123	35.70	-	<0.001

AHI = apnea-hypopnea index on overnight polysomnography, df = degrees of freedom between groups for χ^2^ between groups and within groups for ANOVA, ESS = Epworth Eleepiness Scale; HC = healthy controls, IQ = intelligence quotient, na = not applicable, OSA = obstructive sleep apnea, PSQI = Pittsburg Sleep Quality Index, SaO_2_lowest = lowest oxygen saturation on polysomnography, * = *p* < 0.05, ** = *p* < 0.01, *** = *p* < 0.001 is significantly different compared to HC based on post-hoc analyses, for AHI and SaO_2_lowest, all groups significantly different from each other.

**Table 2 clockssleep-07-00012-t002:** Cognitive complaints in healthy controls (HC) and patients with suspected OSA.

	Range	HC N = 48	AHI < 5 N = 32	AHI 5–15 N = 46	AHI 15–30 N = 25	AHI > 30 N = 24	ANOVA		
							df	F	*p*
CFQ									
Forgetfulness	8–40	18.3 (4.2)	21.9 (5.0)	22.1 (4.4) *	19.8 (5.2)	20.5 (5.2)	4;170	4.97	<0.01
Distractibility	8–40	16.9 (4.0)	20.0 (4.9)	20.4 (4.7) *	18.4 (5.3)	19.4 (4.4)	4;170	4.00	0.01
False triggering	8–40	14.1 (3.0)	16.9 (4.6)	16.3 (4.3)	15.2 (2.)	15.9 (5.1)	4;170	2.73	0.06
BRIEF-A									
Inhibit	8–24	11.8 (2.5)	12.4 (2.7)	12.3 (2.4)	13.0 (3.0)	12.7 (2.4)	4;169	1.30	0.27
Shift	8–24	8.8 (2.3)	9.6 (2.3)	9.7 (2.6)	9.5 (2.6)	10.1 (2.6)	4;169	1.38	0.27
Emotional control	8–24	14.0 (3.9)	15.9 (4.0)	16.5 (3.5)	15.2 (3.6)	16.2 (4.3)	4;169	2.89	0.06
Self-monitor	8–24	8.4 (2.0)	8.7 (2.1)	8.7 (2.3)	9.5 (2.7)	9.3 (1.8)	4;169	1.40	0.27
Initiate	8–24	10.8 (2.4)	11.8 (2.8)	12.8 (2.9)	12.3 (3.6)	14.1 (3.5) ***	4;169	5.93	<0.001
Working memory	8–24	11.3 (2.4)	13.4 (3.4)	14.3 (3.3) ***	13.0 (3.8)	13.9 (3.3)	4;169	6.10	<0.001
Plan/organize	8–24	9.0 (2.1)	14.3 (3.4)	15.3 (3.4)	14.6 (3.3)	15.9 (3.6)	4;169	2.46	0.07
Task monitor	8–24	9.7 (2.3)	9.4 (2.0)	10.1 (2.1)	10.0 (2.4)	10.2 (2.6)	4;169	1.96	0.14
Organization materials	8–24	11.2 (2.7)	12.4 (3.3)	12.5 (3.2)	13.4 (3.7)	13.1 (3.2)	4;169	2.58	0.07

AHI = apnea-hypopnea index; BRIEF-A = Behavior Rating Inventory of Executive Function Adult version; CFQ = Cognitive Failure Questionnaire; df = degrees of freedom between groups and within groups; OSA = obstructive sleep apnea patients; * = *p* < 0.05; *** = *p* < 0.001 is significantly different compared to HC based on post-hoc analyses.

**Table 3 clockssleep-07-00012-t003:** Anxiety, depression, fatigue and performance on neuropsychological tests (norm-corrected Z-scores) in patients with suspected OSA and healthy controls (HC).

	Range	HC N = 48	AHI < 5 N = 32	AHI 5–15 N = 46	AHI 15–30 N = 25	AHI > 30 N = 24	ANOVA		
							df	F	*p*
HADS anxiety	0–21	3.8 (2.6) **	6.4 (2.8) **	6.3 (3.5) **	6.1 (3.5)	5.2 (2.5)	4;169	5.42	<0.001
HADS depression	0–21	3.4 (2.4) **	5.9 (3.7) **	5.9 (3.4) **	5.3 (3.7)	5.2 (3.3)	4;169	4.50	<0.01
FAS fatigue	10–50	20.7 (3.0) ***	26.4 (4.6) ***	26.8 (5.3) ***	25.4 (5.1) **	26.9 (5.2) ***	4;170	13.88	<0.001
*Neuropsychological tests*	Z-scores								
Digit span	M = 0/SD = 1	0.0 (0.9)	0.0 (0.8)	0.0 (1.2)	0.5 (0.9)	0.0 (0.9)	4;170	1.44	0.27
TMT AB-index	M = 0/SD = 1	0.2 (0.8)	0.2 (0.7)	−0.1 (0.9)	0.1 (0.9)	0.1 (1.0)	4;170	1.10	0.40
RAVLT immediate recall	M = 0/SD = 1	−0.2 (1.2) **	−1.2 (0.8) **	−0.6 (1.1)	−0.8 (1.1)	−0.5 (1.1)	4;170	4.50	<0.01
Verbal fluency	M = 0/SD = 1	0.6 (1.1)	0.1 (1.0)	0.1 (1.2)	0.1 (1.2)	−0.1 (0.9)	4;170	2.09	0.13
CNS-VS Reaction time	M = 0/SD = 1	−0.6 (1.1)	−0.9 (1.1)	−1.4 (1.6)	−0.8 (1.6)	−1.3 (1.8)	4;170	2.57	0.09
CNS-VS Processing speed	M = 0/SD = 1	−0.3 (0.9)	−0.6 (0.8)	−0.4 (0.8)	−0.3 (1.1)	−0.3 (0.9)	4;170	0.77	0.55
CNS-VS Complex attention	M = 0/SD = 1	−0.1 (1.2)	−0.4 (1.5)	−0.5 (1.4)	−1.1 (2.1)	−0.8 (1.2)	4;170	2.10	0.13
CNS-VS Cognitive flexibility	M = 0/SD = 1	−0.5 (1,0)	−0.7 (1.2)	−1.0 (1.3)	−1.3 (1.8)	−1.2 (1.2)	4;170	2.01	0.13

CNS-VSs = central nervous system vital signs; df = degrees of freedom between groups and within groups; FAS = Fatigue Assessment Scale; HADS = Hospital Anxiety and Depression Scale; HC = healthy controls; OSA = obstructive sleep apnea; RAVLT = Rey Auditory Verbal Learning Test; TMT = Trail Making Test; ** = *p* < 0.01, *** = *p* < 0.001 are significantly different from HC based on post-hoc analyses.

**Table 4 clockssleep-07-00012-t004:** Complaint-specific analyses of predictors for cognitive complaints. In patients with suspected OSA using subscales of the CFQ.

	CFQ Forgetfulness N = 125 R^2^ = 0.36		CFQ Distractibility N = 125 R^2^ = 0.30		CFQ False Triggering N = 125 R^2^ = 0.25	
	**Stand. ß**	***p*-Value**	**Stand. ß**	***p*-Value**	**Stand. ß**	***p*-Value**
AHI	−0.16	0.29	−0.06	0.81	−0.12	0.53
ESS, sleepiness	0.32	<0.001 ***	0.31	0.01 *	0.34	<0.001 ***
HADS anxiety	0.03	0.86	0.21	0.22	−0.05	0.86
HADS depression	0.05	0.91	0.07	0.86	0.03	0.90
FAS, fatigue	0.32	0.01 *	0.17	0.39	0.22	0.63
Digit Span	0.05	0.86	−0.01	0.95	0.00	0.97
TMT AB-index	0.01	0.95	0.05	0.86	0.09	0.71
RAVLT immediate recall	0.01	0.94	0.08	0.71	0.00	0.97
Verbal fluency	−0.13	0.50	0.00	0.95	−0.14	0.50
CNS-VS Reaction time	0.03	0.93	−0.01	0.95	0.06	0.86
CNS-VS Processing speed	0.02	0.93	−0.03	0.91	0.03	0.90
CNS-VS Complex attention	0.16	0.76	0.16	0.78	0.23	0.65
CNS-VS Cognitive flexibility	−0.09	0.86	0.10	0.86	−0.19	0.76

AHI = Apnea/hypopnea Index; CFQ = cognitive failure questionnaire; CNS-VS = central nervous system-vital signs; ESS = Epworth Sleepiness Scale; FAS = fatigue assessment scale; HADS = hospital anxiety and depression scale; RAVLT =Rey auditory verbal learning test; TMT = trailmaking test; * = *p* < 0.05, *** = *p* < 0.000.

**Table 5 clockssleep-07-00012-t005:** Complaint-specific analyses of predictors for cognitive complaints. In patients with suspected OSA using subscales of the BRIEF-A.

	BRIEF-A Shift N = 124 R^2^ = 0.30		BRIEF-A Emotional Control N = 124 R^2^ = 0.25		BRIEF-A Initiate N = 124 R^2^ = 0.42		BRIEF-A Working Memory N = 124 R^2^ = 0.35		BRIEF-A Plan/Organize N = 124 R^2^ = 0.31		BRIEF-A Task Monitor N = 124 R^2^ = 0.23		BRIEF-A Organize Materials N = 124 R^2^ = 0.24	
	Stand. ß	*p*-Value	Stand. ß	*p*-Value	Stand. ß	*p*-Value	Stand. ß	*p*-Value	Stand. ß	*p*-Value	Stand. ß	*p*-Value	Stand. ß	*p*-Value
AHI	0.12	0.53	−0.02	0.93	0.21	0.07	0.00	0.98	0.15	0.35	0.11	0.64	0.12	0.56
ESS, sleepiness	0.04	0.86	0.10	0.69	0.11	0.58	0.16	0.35	0.08	0.75	0.08	0.76	0.20	0.24
HADS anxiety	0.40	<0.001 ***	0.35	0.01 *	0.08	0.76	0.13	0.56	0.29	0.03 *	0.18	0.36	0.14	0.59
HADS depression	0.07	0.85	−0.13	0.65	0.27	0.04	0.13	0.59	0.03	0.93	0.06	0.86	0.12	0.68
FAS, fatigue	0.04	0.86	0.23	0.19	0.35	<0.001 ***	0.37	<0.001 ***	0.29	0.03 *	0.21	0.26	0.16	0.47
Digit Span	−0.08	0.76	0.05	0.86	0.03	0.86	0.09	0.69	−0.04	0.86	0.06	0.86	0.06	0.86
TMT AB-index	0.03	0.89	−0.07	0.81	0.09	0.69	0.02	0.93	0.04	0.86	−0.04	0.87	−0.08	0.78
RAVLT immediate recall	0.13	0.50	0.10	0.64	0.09	0.66	0.05	0.86	0.08	0.75	0.11	0.64	−0.03	0.90
Verbal fluency	−0.15	0.41	−0.11	0.64	−0.02	0.93	−0.06	0.86	0.02	0.94	0.05	0.86	0.01	0.95
CNS-VS Reaction time	−0.01	0.96	0.07	0.86	0.06	0.86	−0.03	0.90	−0.02	0.95	0.08	0.81	0.08	0.81
CNS-VS Processing speed	−0.14	0.50	0.02	0.94	0.08	0.76	0.02	0.94	−0.07	0.81	0.05	0.86	0.80	0.78
CNS-VS Complex attention	0.08	0.87	0.11	0.86	0.28	0.49	0.25	0.59	0.09	0.86	0.48	0.15	0.37	0.35
CNS-VS Cognitive flexibility	0.08	0.90	−0.11	0.86	−0.24	0.64	−0.24	0.65	0.04	0.94	−0.45	0.28	−0.29	0.59

AHI = apnea-hypopnea Index; BRIEF-A = Behavioral Rating Inventory for Executive Function Adult version; CNS-VSs = central nervous system vital signs; ESS = Epworth Sleepiness Scale; FAS = Fatigue Assessment Scale; HADS = Hospital Anxiety and Depression Scale; RAVLT =Rey Auditory Verbal Learning Test; TMT = Trail Making Test; * = *p* < 0.05; *** = *p* < 0.001; models for BRIEF-inhibit, BRIEF-A self-monitor are not listed because they were not significant (all *p*’s > 0.05).

**Table 6 clockssleep-07-00012-t006:** Total scores on the CFQ and BRIEF-A for healthy controls and patients suspected of OSA with and without final OSA diagnosis.

		Healthy Controls	Suspected OSA				
	Range	N = 48	No OSA Diagnosis N = 38	OSA Diagnosis N = 88	ANOVA		
					df	F	*p*
CFQ	25–125	52.5 (10.4)	60.8 (12.9) **	59.0 (14.7)	2;171	5.03	<0.01
BRIEF-A	75–225	106.6 (19.0)	117.1 (19.7)	120.7 (23.7) **	2;169	6.63	<0.01

BRIEF-A = Behavioral Rating Inventory for Executive Function Adult version; CFQ = Cognitive Failure Questionnaire; df = degrees of freedom between groups and within groups, OSA = obstructive sleep apnea; ** = *p* < 0.01 is significantly different compared to HC based on post-hoc analyses.

**Table 7 clockssleep-07-00012-t007:** Total scores on the CFQ and BRIEF-A for patients with and without cognitive impairments across various OSA severities.

OSA Severity	Cognitive Impairment	N	CFQ	*p*-Value	BRIEF-A	*p*-Value
AHI < 5	yes	11	59.2 (15.8)	0.89	118.2 (24.5)	0.89
	no	21	62.2 (12.8)		116.1 (16.5)	
AHI 5–15	yes	16	62.1 (10.7)	0.90	120.4 (19.7)	0.89
	no	30	61.6 (12.9)		117.3 (27.9)	
AHI 15–30	yes	14	51.4 (16.1)	0.89	113.8 (23.2)	0.59
	no	11	55.5 (20.8)		125.9 (24.1)	
AHI > 30	yes	6	48.0 (7.0)	0.16	114.2 (10.6)	0.59
	no	18	62.2 (13.2)		127.7 (23.0)	

BRIEF-A = Behavioral Rating Inventory for Executive Function Adult version; CFQ = Cognitive Failure Questionnaire; OSA = obstructive sleep apnea.

**Table 8 clockssleep-07-00012-t008:** Content of subscales of CFQ and BRIEF-A.

Subscale	Content
CFQ	
Forgetfulness	Memory slips
Distractibility	Attentional misses, such as absentmindedness
False triggering	Blunders or slips in thinking or motor actions
BRIEF-A	
Inhibit	Problems with controlling impulses
Shift	Problems with thinking flexibly
Emotional control	Problems modulating appropriate emotional responses
Self-monitor	Problems recognizing the effect of one’s behavior on others
Initiate	Problems independently generating ideas
Working memory	Problems holding information in mind to complete a task
Plan/organize	Problems carrying out tasks systematically
Task monitor	Problems assessing mistakes in one’s performance
Organization of materials	Problems keeping one’s workspace in an orderly manner

## Data Availability

The informed consent at the time of the data collection does not allow data to be made available.

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
