# Peer review of "Cognitive Complaints in Patients with Suspected Obstructive Sleep Apnea Are Associated with Sleepiness, Fatigue, and Anxiety, Not with Final Diagnosis or Objective Cognitive Impairment"

_2624-5175, 2025, doi:10.3390/clockssleep7010012_

Round 1
Reviewer 1 Report (Previous Reviewer 3)
Comments and Suggestions for Authors
The authors have nearly completely re-written this paper. The revised manuscript addresses all of my previous concerns. There is a minor error on line 428 where table 6 is referenced, but this should be table 7.
Author Response
- The authors have nearly completely re-written this paper. The revised manuscript addresses all of my previous concerns. There is a minor error on line 428 where table 6 is referenced, but this should be table 7.
We are happy that the revision has been well received and that the issues raised have been addressed. We have corrected the table number from 6 to 7 on line 428 (line 318 in current version).
Reviewer 2 Report (Previous Reviewer 1)
Comments and Suggestions for Authors
The article received is not weel organized as material and methods chapter is at the end of it.
The article still keep long sentences and should be shortened in order to gain accuracy and clarity we recommend shortened it.There are many unclear and overcomplicated sentences
Many sentences are long and convoluted, making it hard for the reader to extract the key points. There are some sections contain excessive jargon without adequate explanation, which can confuse non-specialist readers.
The study measures cognitive complaints using self-reported questionnaires (CFQ, BRIEF-A), which means patients are reporting their own difficulties with memory, attention, and organization and in oly one stage so their are subjected to many biass can affected that solely intake.
Objective cognitive testing is included, but it is limited and does not comprehensively measure all areas of cognitive function.
Control group is based on non OSA patients The non-OSA sleep clinic group includes people with other sleep disorders (e.g., insomnia, restless legs syndrome), which can also affect cognitive complaints. This can biassed the comparison between OSA and non-OSA patient
Auhors should explain different phenotypes according OSA as attention/vigilance dysfunction appears to be associated with sleep fragmentation and global cognitive function with hypoxaemia. In their test they only evaluate AHI but not arousals or RDI.
Please rewrite discussion about limitation of the study in order to provide only AHI and questionnaires results nor other values like PSG or EEG.
"Patients with and without an OSA diagnoses reported more cognitive complaints than HC, with no differences between those with and without cognitive impairments. In conclusion, cognitive complaints were common in subjects suspected of OSA, and were related to anxiety, fatigue and sleepiness rather than objective cognitive performance or impairments". Coclusion needs to be rewritten as is difficult to understand
Author Response
- The article received is not well organized as material and methods chapter is at the end of it.
We followed the guidelines of Clocks & Sleep, which specify that the methods section is to be placed after the discussion. We have carefully reviewed the manuscript and updated the text in several places to ensure the flow, given the required order of sections.
- The article still keep long sentences and should be shortened in order to gain accuracy and clarity we recommend shortened it.There are many unclear and overcomplicated sentences. Many sentences are long and convoluted, making it hard for the reader to extract the key points. There are some sections contain excessive jargon without adequate explanation, which can confuse non-specialist readers.
We have further revised the manuscript throughout, to shorten sentences where possible, and reduced jargon to improve readability.
- The study measures cognitive complaints using self-reported questionnaires (CFQ, BRIEF-A), which means patients are reporting their own difficulties with memory, attention, and organization and in oly one stage so their are subjected to many biass can affected that solely intake. Objective cognitive testing is included, but it is limited and does not comprehensively measure all areas of cognitive function.
The specific focus of the study is on subjective cognitive complaints, which are to be separated from objective impairment. Cognitive complaints were assessed using dedicated and validated self-report questionnaires (CFQ, BRIEF-A) to capture patients’ perceived difficulties in daily life. While objective cognitive testing was included, it was not the primary outcome and was not designed to comprehensively measure all cognitive domains. We acknowledge the limitations of objective cognitive testing and have now further addressed this in the discussion section.
- Control group is based on non-OSA patients The non-OSA sleep clinic group includes people with other sleep disorders (e.g., insomnia, restless legs syndrome), which can also affect cognitive complaints. This can biassed the comparison between OSA and non-OSA patient.
We would like to stress, that in addition to the sleep clinic control group, the study included a healthy control group, recruited outside the sleep clinics. Using both control groups allowed us to explore non-OSA-specific factors related to cognitive complaints, including sleep problems. Advantages and associated considerations of including these two different control groups are now further addressed in the discussion section.
- Auhors should explain different phenotypes according OSA as attention/vigilance dysfunction appears to be associated with sleep fragmentation and global cognitive function with hypoxaemia. In their test they only evaluate AHI but not arousals or RDI. Please rewrite discussion about limitation of the study in order to provide only AHI and questionnaires results nor other values like PSG or EEG.
The influence of mentioned OSA phenotypes is interesting, and could be a topic in future studies, most likely warranting larger datasets, and methods based on the findings of the current study. The discussion has been updated to address the potential role of OSA phenotypes, given the potential influence on objective cognitive dysfunction patterns.
- "Patients with and without an OSA diagnoses reported more cognitive complaints than HC, with no differences between those with and without cognitive impairments. In conclusion, cognitive complaints were common in subjects suspected of OSA, and were related to anxiety, fatigue and sleepiness rather than objective cognitive performance or impairments". Coclusion needs to be rewritten as is difficult to understand.
The conclusion has been revised for clarity:
Cognitive complaints are common among patients seeking evaluation for suspected OSA, regardless of the final diagnosis. In OSA patients, cognitive complaints were linked to subjective sleepiness, anxiety and fatigue. However, the complaints did not reflect objective cognitive impairments. Patients can be reassured that their cognitive concerns are unlikely to indicate true cognitive deficits. When complaints persist, further evaluation and treatment of sleep disturbances, anxiety and fatigue should be considered.
Round 2
Reviewer 2 Report (Previous Reviewer 1)
Comments and Suggestions for Authors
Please reorder material and methods section after introduction.Authors have added succesfully changes suggested by this reviewer.
Author Response
Please reorder material and methods section after introduction.Authors have added succesfully changes suggested by this reviewer.
We are pleased that the revision has been well received. We adhered to the guidelines of Clocks & Sleep, which specify that the methods section should be placed after the discussion.
This manuscript is a resubmission of an earlier submission. The following is a list of the peer review reports and author responses from that submission.
Round 1
Reviewer 1 Report
Comments and Suggestions for Authors
We have to congratulate authors for a well written manuscript but there are severe issues to be addressed,
1) What is the scientific criteria to select patients between <5 and >10 AHI to diagnose from OSA. ? This has no sense as <15 are considered mild sleep apnea and both categories are included in the selection. Starting from this point all the investigation is biassed. We recommed to select patients form AHI >30 and reevaluate the findings published. We recommend author read the possibility of missdiagnosssis with only one night .A 2022 report examining the validity of on night at-home sleep studies found an overall misdiagnosis rate of 39%. Massie F, Van Pee B, Bergmann J. Correlations Between Home Sleep Apnea Tests and Polysomnography Outcomes Do Not Fully Reflect the Diagnostic Accuracy of These Tests
Reviewer 2 Report
Comments and Suggestions for Authors
Manuscript ID: clockssleep-3291451
Review Summary
The manuscript explores the cognitive complaints in OSA and aims to identify potential predictors of cognitive complaints in these patients. The practical implications for clinical practice are well articulated especially in advising clinicians to consider factors beyond OSA when addressing cognitive complaints in patients.
While the manuscript provides valuable insights and contributes to understanding the complexities of cognitive function in OSA, it requires major revisions and clarifications on the following items. In addition, overall readability of the manuscript could be enhanced by restructuring and breaking long sentences into shorter ones. I believe addressing these concerns will strengthen the manuscript and provide a clearer foundation for the study.
Introduction:
· While the manuscript defines "cognitive complaints" as concerns about everyday cognitive abilities (lines 43-44), a more precise operational definition or description is needed. What exactly constitutes a cognitive complaint? Clarifying this in the introduction would ensure that readers understand how cognitive complaints are being measured.
· The section discussing cognitive complaints (lines 50-64) and their potential correlates could be more tightly integrated. It feels somewhat disconnected from previous discussion of objective cognitive functioning in OSA. A smoother transition between objective findings and discussion of subjective complaints would improve the flow.
· The introduction suggests a potential link between cognitive complaints and symptoms such as sleepiness, fatigue, anxiety and depression. However, the literature on this topic is not discussed in enough detail. This is an important gap in literature that warrants more discussion.
· The control groups in this study include OSA patients, clinic controls and healthy community controls. The eligibility criteria are described in methods section, however, more justification is needed for the inclusion of clinic controls, particularly those referred for symptoms that resemble OSA such as sleepiness, depression, and fatigue. It would be helpful to explain why these individuals were chosen as controls and how their characteristics compare to OSA patients in terms of cognitive complaints. This information could be included in the results section. Also, the inclusion of hospital staff and their families as a healthy control group should be justified to how do they represent the general population in terms of cognitive complaints and are there potential biases in their selection?
· The hypotheses are clearly stated, but further detail is needed regarding the reasoning behind these predictions. For instance, why do the authors hypothesize that OSA patients will report more cognitive complaints than healthy controls but fewer than clinic controls? Providing more rationale for these predictions would enhance the transparency of the study design.
Results:
· Details of power analysis or sample size calculations are needed as the authors mention a small sample as a limitation of the study. Also, including effect sizes would help contextualize the magnitude of the findings of the study.
· A rationale for not including or characterizing mild OSA (AHI = 5-10) patients is needed.
· The numbers in the text and Figure 1 do not align correctly. For example, 127 minus 32 equals 95, not 81, as shown in the flowchart in Figure 1. Also, the text indicates that there were 24 clinic controls, while the figure states that there were 25. Please review the sample sizes.
· Since participants with missing data were excluded, discussion of the potential bias introduced by this exclusion is important.
· Please clarify what the p-values in Tables 1, 2 and 3 represent, particularly those to the right of "healthy community controls," as their meaning is unclear, i.e., whether these p-values represent the differences between the clinic controls or the OSA groups with the healthy community controls. Similarly, Table 5 needs clarification in terms of meanings of p-values.
Statistical Methods:
· The authors state that participants were matched for age, sex, education, and IQ. However, it would be valuable to report on how successful this matching was and whether any residual imbalances were statistically tested (e.g., by comparing the means or distributions of these variables).
· The use of ANOVA for group comparisons and t-tests for post-hoc analysis is standard practice. However, the report should include the F-statistics and degrees of freedom for ANOVAs to provide more detail on the findings. Including ANOVA outputs will help address this concern.
· The Benjamini-Hochberg correction for multiple comparisons is appropriate and should be consistently applied across all statstical analysis. The manuscript should present the corrected p-values to ensure transparency.
Discussion:
· The discussion repeatedly emphasizes associations between subjective factors (like sleepiness, fatigue, anxiety) and cognitive complaints but does not sufficiently explore potential directionality of these relationships. For instance, it remains unclear whether cognitive complaints might exacerbate fatigue and anxiety, or if these factors (fatigue, anxiety, etc.) lead to more pronounced cognitive complaints. A deeper exploration of the bidirectional nature of these relationships would add complexity to the interpretation of findings.
· While the study observes no significant relationship between OSA severity (AHI) and cognitive complaints for most domains, a discussion on why AHI is linked only to complaints about "initiative" could be expanded.
· Although the authors suggest that cognitive complaints in OSA are not related to cognitive performance, this conclusion may require more nuanced discussion. The study's reliance on "norm-corrected Z-scores" for neuropsychological tests, without further validation of the clinical relevance of these norms, may raise concerns. While it is true that OSA group performed comparably to healthy controls in most neuropsychological tests, the finding that OSA patients performed worse in verbal fluency raises an important question about the specificity of cognitive complaints in OSA. Exploring this in more detail would help clarify whether cognitive complaints reflect more subtle cognitive impairments that are not captured by the current neuropsychological testing battery. Including this information could enhance the discussion's depth.
· The authors mention that relatively small sample size and an absence of a substantial number of female participants could limit the generalizability of their findings. While they note this limitation, it would be helpful if they could provide more insight into how this limitation might specifically affect their conclusions. For example, the separate gender analyses could provide understanding whether cognitive complaints and performance differ between male and female patients.
· The finding that around 40% of OSA patients are classified as cognitively impaired is noteworthy. The authors mention that cognitive complaints are not related to objective cognitive impairments, yet a significant subset of OSA patients is identified as cognitively impaired. It would be valuable to discuss whether the subjective nature of cognitive complaints in OSA could reflect a more complex interplay of factors such as sleep disruption, comorbidities, or unrecognized cognitive impairments. This could help reconcile the discrepancies between subjective complaints and objective neuropsychological assessments of cognition in OSA.
Minor Edits
· Please move the methods section before the results for a more comprehensive flow.
· Please do not use abbreviations when a term appears first time in the text. For example, PSQI, CFQ, FAS etc. Please review all abbreviations.
· Please restructure lines 25-28 in the abstract for a clearer understanding of the concluding sentence.
· Please provide a reference for lines 54-56 - “Since cognitive complaints do not seem to correlate with objective measures…”
· Please write P values as 0.01 and not .01.

The manuscript could benefit from more concise phrasing, restructuring sections for better coherence and carefully reviewing the text for redundancy.
Reviewer 3 Report
Comments and Suggestions for Authors
In this study Vasessen et al compared OSA patients recruited from OSA clinics of two hospitals, to patients referred to the clinics without OSA as well as a group of community controls to determine the prevalence of cognitive complaints among these groups. They found cognitive complaints to be more common among OSA patients than community controls and that these complaints were variously associated with sleep apnea severity, sleepiness, anxiety and fatigue, but not objective cognitive function. OSA patients without cognitive impairment had more cognitive complaints than those with OSA and community and OSA clinic controls.
The paper is clearly written and organized. The bottom line appears to be that cognitive complaints in patients presenting to a sleep clinic for possible OSA are not associated with a diagnosis of OSA and are not associated with cognitive performance as well; the authors suggest that cognitive complaints in this population just may be a characteristic of a population seeking care for OSA. The authors should address the following issues:
1. OSA was defined as AHI >10 and No OSA as AHI <5. There is no justification as to why this was done. The “cutpoint” for the presence of OSA is generally considered as >5.
2. Lowest SaO2 was used as predictor. Cumulative time below 90% saturation would be a better marker of the potential impact of hypoxemia.
3. The authors indicate they had insufficient power to include all predictors into a single regression. Did they perform a power analysis before starting the study to determine the number of participants needed for the study?
4. Cognitive impairment was defined as one or more abnormal scores on neuropsychological testing. Why was this cutpoint employed?
5. Did the authors evaluate the effect of race/ethnicity? If not, this should be a limitation because cognitive complaints inasmuch as some data indicate that non-white populations have more cognitive complaints than white populations.
Some minor issues are listed below:
1. Line 77: “Our clinic controls were referred by a general practitioner…” However, the Methods on lines 269-272 indicate that patients were recruited from the OSA units of two general hospital and only were divided post-hoc into OSA patients and non OSA clinic controls. Please clarify this.
2. “therefor” is misspelled throughout the paper. It should be ‘therefore’.
3. Line 142 discusses “verbal fluency”. However, Table 3 lists “animal fluency”. Please clarify.